# Spike residue 403 affects binding of coronavirus spikes to human ACE2

Fabian Zech[1], Daniel Schniertshauer[1], Christoph Jung[2,3,4], Alexandra Herrmann[5], Arne Cordsmeier[5], Qinya Xie[1], Rayhane Nchioua[1], Caterina Prelli Bozzo[1], Meta Volcic [1], Lennart Koepke [1], Janis A. Müller [1], Jana Krüger [6], Sandra Heller [6], Steffen Stenger[7], Markus Hoffmann [8], Stefan Pöhlmann [8], Alexander Kleger [6], Timo Jacob[2,3,4], Karl-Klaus Conzelmann [9], Armin Ensser[5], Konstantin M. J. Sparrer [1] & Frank Kirchhoff [1✉]

The bat sarbecovirus RaTG13 is a close relative of SARS-CoV-2, the cause of the COVID-19 pandemic. However, this bat virus was most likely unable to directly infect humans since its Spike (S) protein does not interact efficiently with the human ACE2 receptor. Here, we show that a single T403R mutation increases binding of RaTG13 S to human ACE2 and allows VSV pseudoparticle infection of human lung cells and intestinal organoids. Conversely, mutation of R403T in the SARS-CoV-2 S reduces pseudoparticle infection and viral replication. The T403R RaTG13 S is neutralized by sera from individuals vaccinated against COVID-19 indicating that vaccination might protect against future zoonoses. Our data suggest that a positively charged amino acid at position 403 in the S protein is critical for efficient utilization of human ACE2 by S proteins of bat coronaviruses. This finding could help to better predict the zoonotic potential of animal coronaviruses.

[1] Institute of Molecular Virology, Ulm University Medical Center, 89081 Ulm, Germany. [2] Institute of Electrochemistry, Ulm University, 89081 Ulm, Germany. [3] Helmholtz-Institute Ulm (HIU) Electrochemical Energy Storage, Helmholtz-Straße 16, 89081 Ulm, Germany. [4] Karlsruhe Institute of Technology (KIT), P.O. Box 3640, 76021 Karlsruhe, Germany. [5] Institute of Clinical and Molecular Virology, University Hospital Erlangen, Friedrich-Alexander Universität Erlangen-Nürnberg, 91054 Erlangen, Germany. [6] Department of Internal Medicine I, Ulm University Medical Center, 89081 Ulm, Germany. [7] Institute of Medical Microbiology and Hygiene, Ulm University Medical Centre, 89081 Ulm, Germany. [8] Infection Biology Unit, German Primate Center—Leibniz Institute for Primate Research, Göttingen, Germany. [9] Max von Pettenkofer-Institute of Virology, Medical Faculty, and Gene Center, Ludwig-Maximilians-Universität München, 81377 Munich, Germany. ✉email: Frank.Kirchhoff@uni-ulm.de

Since its first occurrence in December 2019, SARS-CoV-2, the causative agent of COVID-19, has infected more than 220 million people by September 2021 and caused a global health and economic crisis[1,2]. SARS-CoV-2 belongs to the *Sarbecovirus* subgenus of betacoronaviruses, which are mainly found in bats[3,4]. Horseshoe bats (*Rhinolophidae)* also harbour viruses that are closely related to SARS-CoV-1 that infected about 8.000 people in 2002 and 2003[3,4]. The bat virus RaTG13 sampled from a *Rhinolophus affinis* horseshoe bat in 2013 in Yunnan has been identified as one of the closest relatives of SARS-CoV-2, showing approximately 96% sequence identity throughout the genome[1]. Thus, SARS-CoV-2 most likely originated from horseshoe bats[1,2], although it has been proposed that cross-species transmission to humans may have involved pangolins as secondary intermediate host[5,6].

The Spike (S) proteins of both SARS-CoV-1 and SARS-CoV-2 utilize the angiotensin-converting enzyme 2 (ACE2) receptor to enter human target cells[7–10]. The ability to use a human receptor for efficient infection is a key prerequisite for successful zoonotic transmission. Although the RaTG13 S protein is highly similar to the SARS-CoV-2 S it does not interact efficiently with the human ACE2 receptor[11], suggesting that this bat virus would most likely not be able to infect humans directly. It has been reported that specific alterations in the receptor-binding domain (RBD)[12], as well as a four-amino-acid insertion (PRRA) introducing a furin-cleavage site[7,13], play a key role in efficient ACE2 utilization and consequently the high infectiousness and efficient spread of SARS-CoV-2. Here we show, which specific features allow the S proteins of bat CoVs to use human ACE2 for efficient entry and thus to successfully cross the species barrier to humans.

## Results

**Computational analysis suggests that R403 contributes to Spike interaction with human ACE2.** Previous simulations suggested that R403 is involved in intramolecular interactions stabilizing the SARS-CoV-2 S trimer interface[11] and contributes significantly to the strength of SARS-CoV-2 RBD interaction with the human ACE2 receptor[14–16]. We found that R403 is highly conserved in SARS-CoV-2 S proteins: only 294 of ~3.4 million S sequences recorded on GSAID contain a conservative change of R403K and just 132 another amino acids: M (78), H (16), G (10), S (10), T (6), I (9), L (2), N (2), P (1) or W (1). Once residue 403 was deleted. Notably, the presence of a positively charged residue at position 403 distinguishes the S proteins of SARS-CoV-1 (K403) and SARS-CoV-2 (R403) from the bat CoV RaTG13 S protein (T403) (Fig. 1a). Molecular modelling of S/ACE2 interaction using reactive force field simulations confirmed the establishment of close proximity and putative charge interactions between R403 in the SARS-CoV-2 S with E37 in the human ACE2 receptor (Fig. 1b). These analyses predicted that mutation of T403R significantly strengthens the ability of the RaTG13 S protein to bind human ACE2 (Fig. 1c and Supplementary Movies 1 and 2).

**Mutation of T403R allows RaTG13 S to use human ACE2 as entry receptor.** To verify the functional importance of residue 403 for ACE2 usage by CoV S proteins, we used VSV particles (VSVpp) pseudotyped with parental and mutant S proteins. Mutation of R403T reduced the ability of the SARS-CoV-2 S protein to mediate entry of VSVpp into the human colorectal adenocarcinoma cell line Caco-2 by 40% (Fig. 2a). Strikingly, the T403R change enhanced the infectiousness of Vesicular stomatitis virus pseudoparticles (VSVpp) carrying the RaTG13 S for Caco-2 cells ~30-fold, while substitution of T403A introduced as control had no enhancing effect (Fig. 2a). Similar results were obtained in

the human lung cancer cell line Calu-3 (Fig. 2b) and the lung carcinoma cell line A549 that required overexpression of ACE2 (Fig. 2c; Supplementary Fig. 1). Coexpression of the transmembrane serine protease 2 (TMPRSS2) enhanced infection mediated by the wildtype (WT) and R403T SARS-CoV-2 S proteins but had no significant effect on entry mediated by the RaTG13 T403R S (Fig. 2c). To assess whether the T403R change might allow the bat CoV RaTG13 to spread to human bs, we performed infection studies using intestinal organoids derived from pluripotent stem cells. The parental SARS-CoV-2 S protein allowed efficient infection of gut organoids[17] and the R403T change had modest attenuating effects (Fig. 2d and Supplementary Fig. 2). In contrast, the parental RaTG13 S protein did not mediate VSVpp entry, while the T403R S allowed significant infection of human intestinal cells (Fig. 2d; Supplementary Fig. 2).

Cell-to-cell fusion assays showed that coexpression of the SARS-CoV-2 S and human ACE2 resulted in the formation of large syncytia (Supplementary Fig. 3). The parental and T403A RaTG13 S did not lead to detectable fusion. However, significant syncytia formation was observed for the T403R RaTG13 S (Supplementary Fig. 3). In line with the VSVpp results, complementation of a full-length recombinant SARS-CoV-2 lacking the S ORF (SCoV-2ΔS, replicon System) in ACE2-expressing HEK293T cells with WT SARS-CoV-2 S led to virus-induced cytopathic effects (CPE) indicating successful virus production and propagation (Fig. 2e). Mutation of R403T in the SARS-CoV-2 S reduced CPE. The WT and T403A RaTG13 S were unable to complement SCoV-2ΔS, while the T403R RaTG13 S resulted in significant CPE. Expression of a Gaussia luciferase (GLuc) from S variant complemented recombinant SCoV2ΔS-GLuc confirmed the importance of R403 for viral spread (Fig. 2f). To assess the relevance of R403 for SARS-CoV-2 replication, we reconstituted replication-competent SARS-CoV-2 using bacmids specifically containing the R403T mutation and coding for the yellow fluorescent protein (YFP) in place of ORF6[18]. R403T S was efficiently expressed and incorporated into viral particles, albeit at slightly reduced levels compared to the WT SARS-CoV-2 S (Supplementary Fig. 4). Infection of Caco-2 cells at low MOI showed that the R403T SARS-CoV-2 mutant replicated with significantly lower efficiency than WT virus (Fig. 2g). Thus, data obtained with replication-competent SARS-CoV-2 confirmed the results of S-mediated VSVpp infection and the replicon assays.

**The enhancing effect of T403R in RaTG13 S depends on E37 in ACE2.** Modelling analyses suggest that R403 promotes S-mediated infection of human cells because it interacts with E37 in human ACE2 (Fig. 1b). In agreement with the in silico data, mutation of E37A in ACE2 abolished the enhancing effect of the T403R change on RaTG13 S-mediated VSVpp infection of HEK293T cells (Fig. 3a), without affecting ACE2 expression levels (Fig. 3b). In addition, the substitution of E37A reduced the levels of WT SARS-CoV-2 S-mediated infection to those mediated by R403T SARS-CoV-2 S (Fig. 3a). In comparison, mutation of D38A in ACE2 predicted to play little if any role in S interaction did not disrupt the enhancing effect of the T403R change on RaTG13 S-mediated VSVpp infection.

To examine the role of R403 in the S protein on ACE2 binding, we established an in vitro S-ACE2 interaction assay. Immobilized ACE2 is incubated with lysates of S-V5 transfected HEK293T cells (Fig. 3c). S protein retained after washing is detected by an αV5-Ms and quantified using a secondary HRP-conjugated anti-mouse Ab. SARS-CoV-2 S was expressed at lower levels compared to the RaTG13 S (Fig. 3d) but bound efficiently to human ACE2 (Fig. 3e). The R403T change in SARS-CoV-2 S moderately reduced and the T403R substitution in RaTG13 S

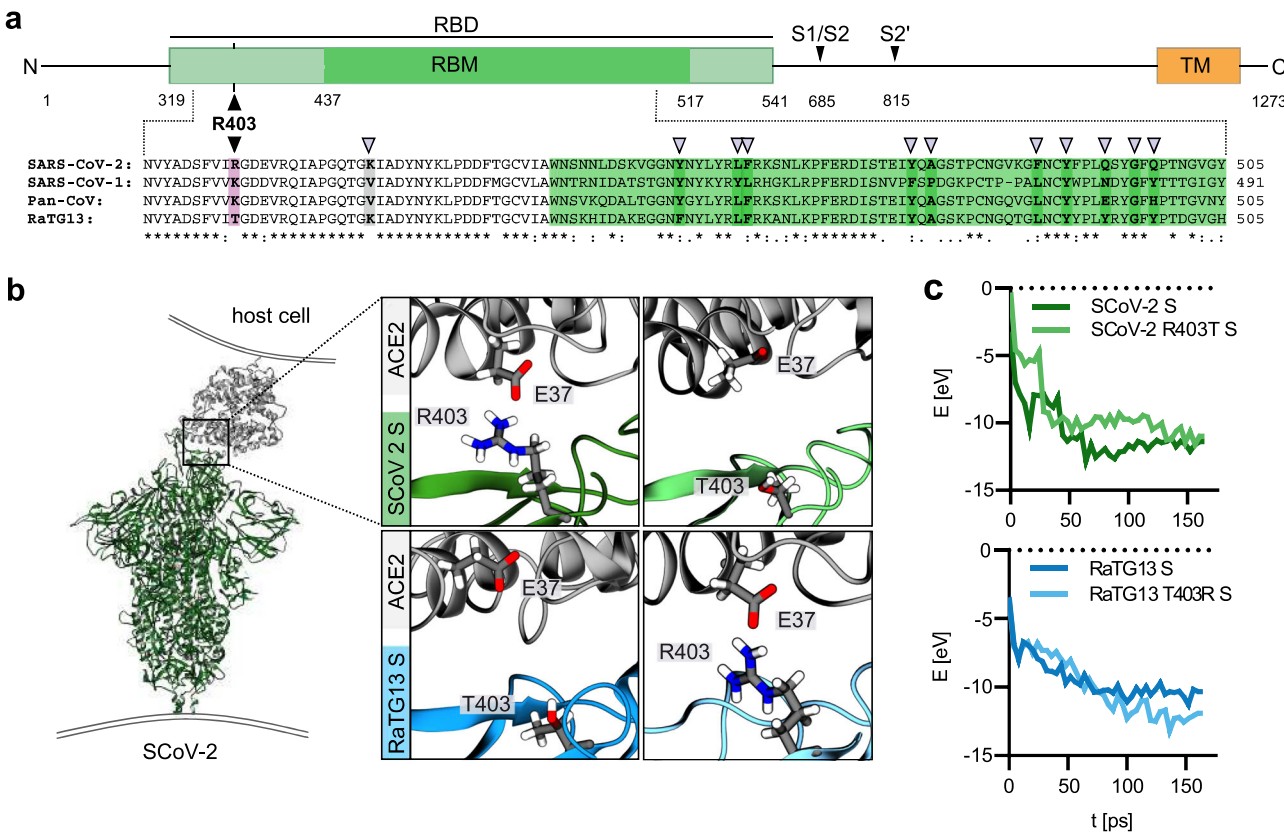

**Fig. 1 Modelling of the interaction of Coronavirus Spike residue 403 with human ACE2. a** Schematic representation of the SARS-CoV-2 S protein (top panel), domains are indicated in different colours. Receptor binding domain (RBD), light green. Receptor binding motif (RBM), dark green. Transmembrane domain (TM), orange. R403, pink. S1/S2 and S2′ cleavage sites are indicated. Sequence alignment of SARS-CoV-2, SARS-CoV-1, Pan-CoV, and RaTG13 Spike RBD (bottom panel). Sequence conservation is indicated. grey arrows denote important residues for ACE2 binding. **b** Reactive force field simulation of SARS-CoV-2 Spike in complex with human ACE2 (PDB: 7KNB (https://www.rcsb.org/structure/7KNB)) (left panel) and focus on position 403 in SARS-CoV-2 S (R) or RaTG13 S (T) or respective exchange mutants at position 403 (right panel). **c** Exemplary energy curve of the reactive molecular dynamics simulation for SARS-CoV-2 S and SARS-CoV-2 S R403T (top panel) and RaTG13 and RaTG13 T430R spike with human ACE2 (bottom panel).

strongly enhanced the levels of S protein bound to ACE2 (Fig. 3e). In the case of SARS-CoV-2 R403T S this effect may be partly due to reduced S expression levels. The T403R change in RaTG13 S, however, specifically enhanced ACE2 binding without affecting S expression levels.

**Proteolytic processing of the T403R RaTG13 S protein.** Coronavirus entry is a multi-step process and critically dependent on proteolytic processing of the S protein[19]. It has been reported that sequential cleavage of SARS-CoV-2 S first by furin at the S1/S2 site and subsequently by TMPRSS2 at the S2′ site is required for efficient infection[9,20]. The RaTG13 S lacks the poly-basic furin cleavage site which contributes to the high infectivity of SARS-CoV-2, while the presumed TMPRSS2 cleavage site is conserved (Fig. 4a). Western blot analyses showed that the mutant S proteins were efficiently expressed and incorporated into VSVpp (Fig. 4b). Predictably, cleavage at the S1/S2 site was less efficient for RaTG13 compared to the SARS-CoV-2 S proteins (Fig. 4b). SARS-CoV-2 replicates in ACE2 expressing cells and it has been reported that the interaction of the S protein with ACE2 promotes proteolytic processing[21,22]. Indeed, ACE2 coexpression induced processing of the WT and R403T SARS-CoV-2 as well as T403R RaTG13 S2 proteins to S2′, while cleavage of the WT and T403A RaTG13 S proteins remained inefficient (Fig. 4b).

R403 generates a potential RGD integrin-binding site in the viral Spike protein and it is under debate whether the ability of

the SARS-CoV-2 S to use integrins as viral attachment factors may play a role in its high infectiousness[23–25]. We found that the integrin inhibitor ATN-161 had no definitive effect on SARS-CoV-2 or T403R RaTG13 S-mediated infection of Caco-2 cells (Supplementary Fig. 5) reported to express α5β1 Integrin[26]. Thus, the enhancing effect of the T403R mutation on the ability of RaTG13 S to infect human cells seems to be due to increased interaction with ACE2 rather than the utilization of integrins.

RaTG13 S-mediated VSVpp infection of A549 cells was equally effective in the presence or absence of TMPRSS2 (Fig. 2c). It has been reported that the cysteine proteases Cathepsin B/L can activate S in the absence of TMPRSS2[27,28]. Indeed, SARS-CoV-2 and T403R RaTG13 S-mediated VSVpp infection of ACE2-A549 cells was efficiently inhibited by the Cathepsin inhibitor E64-d but not by the TMPRSS2 inhibitor Camostat mesylate (Fig. 4c). Thus, activation of S2 to S2′ during RaTG13 T403R-dependent infection of ACE2-A549 cells is mediated by Cathepsins. To assess which proteases are involved in cell−cell fusion mediated by the RaTG13 T403R S protein, we examined syncitia formation in the presence of specific inhibitors. Furin inhibitor 1 prevented and the Cathepsin inhibitor E64-d moderately reduced both SARS-CoV-2 and RaTG13 T403R S-mediated cell−cell fusion (Supplementary Fig. 6). In contrast, the TMPRSS2 inhibitor Camostat had little if any inhibitory effect. Altogether, mutation of T403R allows RaTG13 S interaction with human ACE2 and

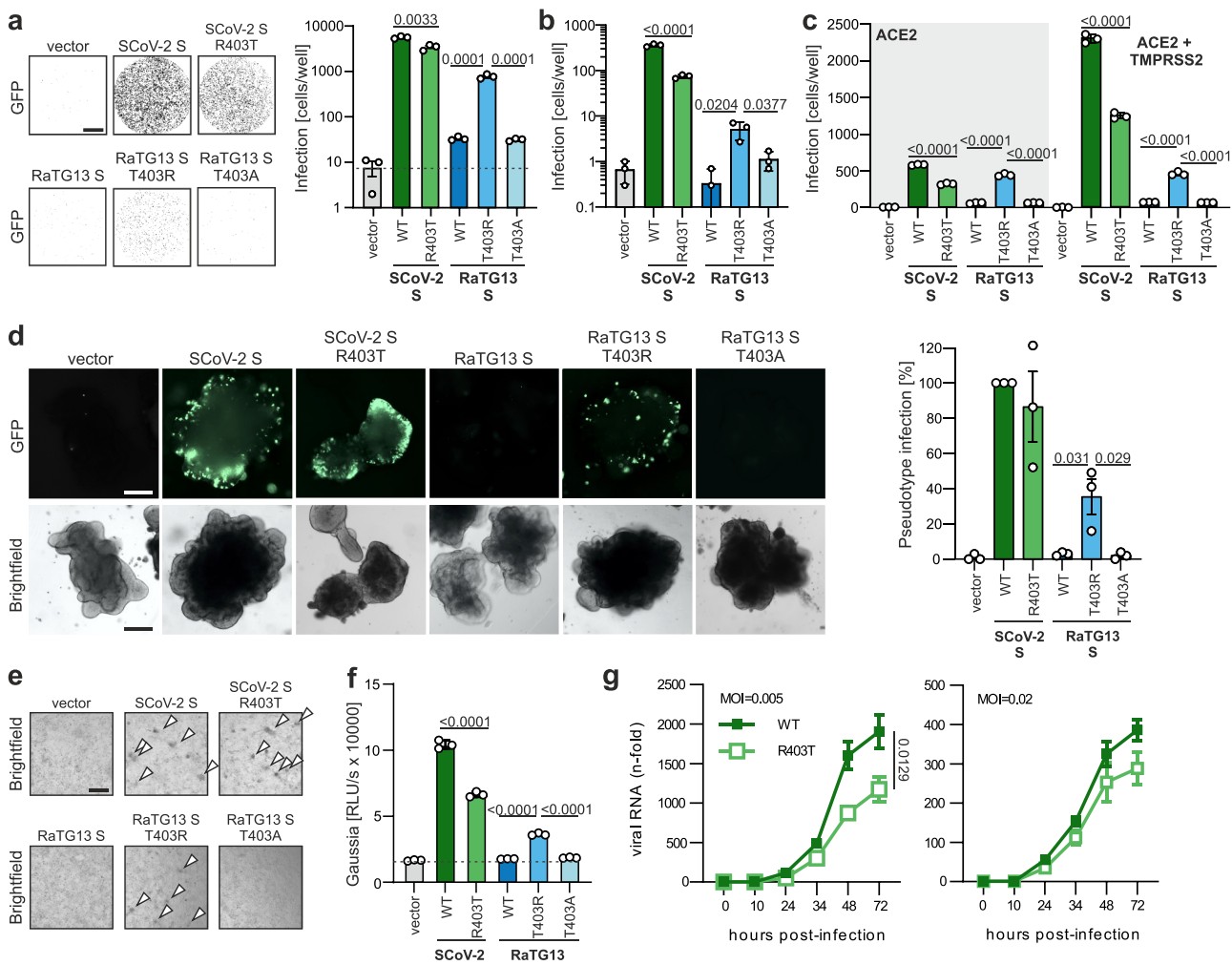

**Fig. 2 R403 in Spike is crucial for ACE2-dependent entry. a** Binary images of CaCo-2 cells transduced with VSVΔG-GFP pseudotyped with SARS-CoV-2, RaTG13 or indicated mutant S. Successful infection events (=GFP positive cells) displayed as black dots. Scale bar, 1.5 mm and automatic quantification of infection events by counting GFP positive cells. Bars represent the mean of three independent experiments (±SEM); **b** automatic quantification of infection events of Calu-3 cells transduced with VSVΔG-GFP pseudotyped with SARS-CoV-2, RaTG13 or indicated mutant S. Bars represent the mean of three independent experiments (±SEM). **c** Automatic quantification of infection events of A459 ACE2 and A459 ACE2 and TMPRSS2 expressing cells transduced with VSVΔG-GFP pseudotyped with SARS-CoV-2, RaTG13 or indicated mutant S. Bars represent the mean of three independent experiments (±SEM). **d** Bright field and fluorescence microscopy (GFP) images of hPSC derived gut organoids infected with VSVΔG-GFP (green) pseudotyped with SARS-CoV-2, RaTG13, or indicated mutant S (300 µl, 2 h). Scale bar, 250 µm and quantification of the percentage of GFP-positive cells of (**a**). Bars represent the mean of three independent experiments (±SEM). **e** Bright field and fluorescence microscopy (GFP) images of HEK293T cells transfected with SCoV-2ΔS bacmid, SCoV2-N, ACE2, T7 polymerase and indicated Spike variants. Scale bar, 125 µm. Arrows indicate syncytia. **f** Quantification of Gaussia luciferase activity in the supernatant of HEK293T cells expressing SCoV-2ΔS-Gaussia bacmids as described in (**c**). Bars represent the mean of three independent experiments (±SEM). **g** Replication kinetic of Caco-2 cells, infected with either SARS-CoV-2 d6-YFP wild type or SARS-CoV-2 d6-YFP R403T (MOI 0.005 or 0.02). Supernatants were collected at the indicated time points post-infection, and replication was determined by RT-qPCR. Lines represent the mean of three independent experiments (±SEM). **a–d**, **f** Two-tailed Student's t-test with Welch's correction. **g** Two-tailed Student's t-test with Welch's correction (on area under the curve).

proteolytic activation by furin and Cathepsins for both cell–cell fusion and RaTG13 S-mediated VSVpp entry.

**The ability of T403R RaTG13 S to utilize ACE2 is species-specific.** To examine the species-specificity of receptor usage by SARS-CoV-2 and RaTG13 S proteins, we overexpressed human and bat-derived ACE2 in HEK293T cells and examined their susceptibility to S-mediated VSVpp infection. The WT SARS-CoV-2 and the T403R RaTG13 S proteins allowed efficient entry into cells overexpressing human ACE2, while the parental RaTG13 S protein was poorly active (Fig. 5a). Both WT SARS-CoV-2 S and (to a lesser extent) R403T SARS-CoV-2 S proteins

were also capable of using bat (*Rhinolophus affinis*) ACE2 for viral entry although the overall infection rates were low (Fig. 5a). In contrast, the WT RaTG13 S protein used *R. pusillus* ACE2 with very poor efficiency and was unable to use *R. macrotis* ACE2 for infection, suggesting that RaTG13 might use an alternative receptor for infection of bat cells. The results agree with the previous finding that RaTG13 S is able to use human ACE2 to some extent if overexpressed[29] but also confirm that the T403R change greatly enhances this function and is required for utilization of endogenously expressed human ACE2. To validate the results obtained with human HEK293T cells, we utilized the lung epithelial cell line Tb1 Lu1 of *Tadarida brasiliensis* (Bat31)[30]. In

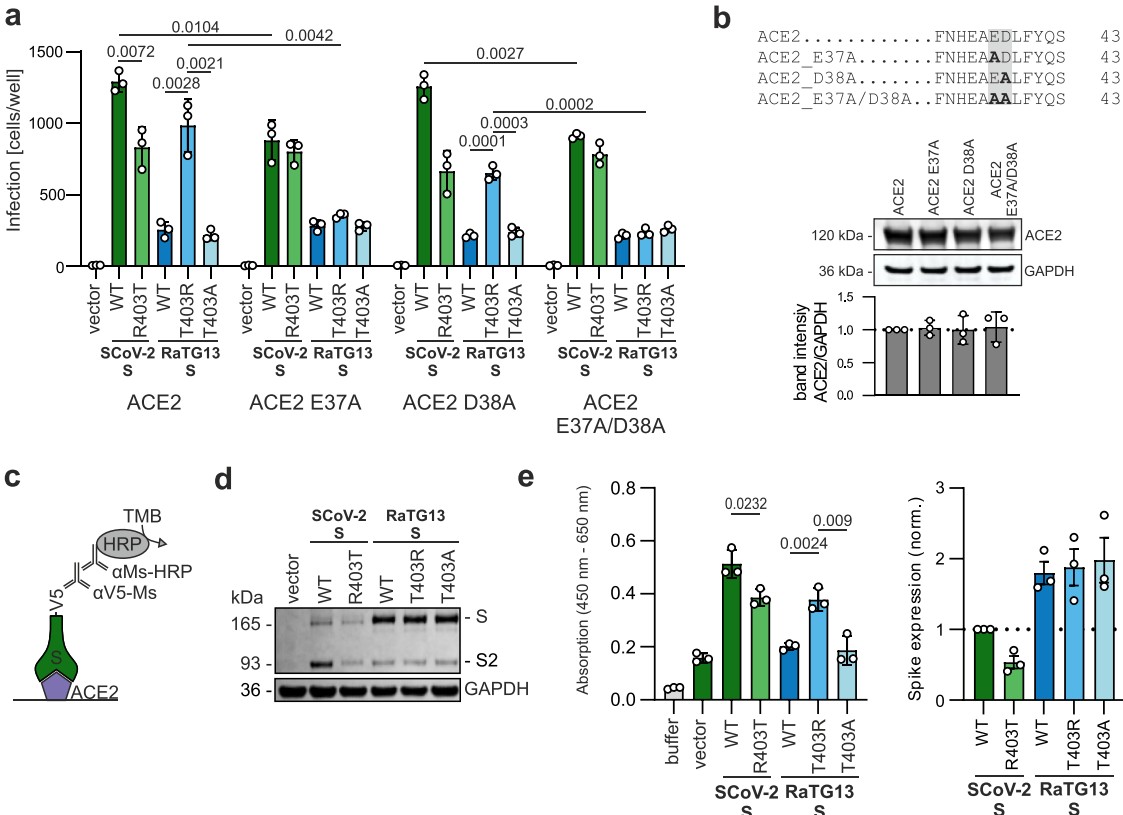

**Fig. 3 The enhancing effect of T403R in RaTG13 S depends on E37 in ACE2. a** HEK293T cells expressing indicated ACE2 constructs were infected with VSVΔG-GFP pseudotyped with SCoV-2, RaTG13, or indicated mutant S. Quantification by automatic counting of GFP positive cells. Bars represent the mean of three independent experiments (±SEM). Two-tailed Student's t-test with Welch's correction. **b** Mutations introduced into human ACE2 (upper) and exemplary immunoblot of whole cells lysates (WCLs) of HEK293T cells expressing the indicated ACE2 constructs. Blots were stained with anti-ACE2 and anti-GAPDH and quantified for ACE2 expression (lower panel). Bars represent the mean of three independent experiments (±SEM). **c** Schematic representation of the ACE2 interaction assay. **d** Exemplary immunoblots of WCLs of HEK293T cells expressing SCoV-2 S, RaTG13 S, or the indicated mutant. Blots were stained with anti-SCoV-2 S, anti-GAPDH, and quantified for Spike expression. **e** ACE2 binding using whole cell lysates of HEK293T expressing SARS-CoV-2, RaTG13, or indicated mutant S. Bars represent the mean of three independent experiments (±SEM).

agreement with the previous finding that this cell line lacks endogenous ACE2 expression, it did not support infection by CoV S proteins (Fig. 5b). Engineered expression of human ACE2 rendered Lu 1 highly susceptible to infection mediated by SARS-CoV-2 and the T403R RaTG13 S proteins (Fig. 5b). In comparison, entry via the R403T SARS-CoV-2 S was strongly attenuated and the WT and T403A RaTG13 S proteins were unable to mediate VSV-pp infection.

**Sensitivity of T403R RaTG13 S to inhibitors and sera from vaccinated individuals**. The increased infectiousness of the T403R RaTG13 S enabled us to examine its sensitivity to therapeutic agents and serum neutralization. In agreement with its reported broad antiviral activity[31], the fusion inhibitor EK-1 efficiently inhibited SARS-CoV-2, T403R RaTG13, as well as (to a much lesser extent) Pangolin CoV S-mediated infection (Fig. 6a). In contrast, the monoclonal antibody Casirivirab[32] was only active against SARS-CoV-2 S (Fig. 6b). Sera from three individuals who had received prime-boost treatment with the Pfizer-BioNTech COVID-19 Vaccine neutralized the T403R RaTG13 S with higher efficiency than the SARS-CoV-2 S (Fig. 6c). To further analyze the sensitivity of the T403R RaTG13 S to neutralization, we examined sera from 22 individuals who received heterologous ChAdOx1 nCoV-19/BNT162b2 prime-boost vaccination and nine individuals who received homologous BNT162b2 vaccination. In contrast to the initial experiment (Fig. 6c), VSVpp

stocks were normalized for the same infectivity. We found that all sera inhibited RaTG13 T403R S-mediated infection abeit with varying efficiency (Fig. 6d). On average, sera obtained after heterologous and homologous vaccination regimens showed similar neutralization efficiencies against RaTG13 (Fig. 6e) and SARS-CoV-2 (Fig. 6f). These results suggest that current vaccine regimens against SARS-CoV-2 may protect against future zoonoses of bat coronaviruses.

## Discussion

Our results demonstrate that a single amino acid change of T403R allows the S protein of RaTG13, one of the closest known bat relatives of SARS-CoV-2, to utilize human ACE2 for viral entry. The strong enhancing effect of the T403R change on RaTG13 S function came as a surprise since five of six different residues proposed to be critical for SARS-CoV-2 S RBD interaction with human ACE2 are not conserved in RaTG13 S[12,33]. We show that the effect of T403R on RaTG13 S is due to an increased ability to interact with E37 on human ACE2, allowing more efficient receptor binding. Mutation of E37A not only abolished the enhancing effect of the T403R change on RaTG13 S-mediated VSVpp infection but also reduced infection mediated by the SARS-CoV-2 S (Fig. 3a). Some individuals show rare polymorphisms of E37K (frequency: 3.27e−5; gnomAD, https:// gnomad.broadinstitute.org) in ACE2 which was reported to impair S binding[34], similar to the E37A mutant analyzed in the

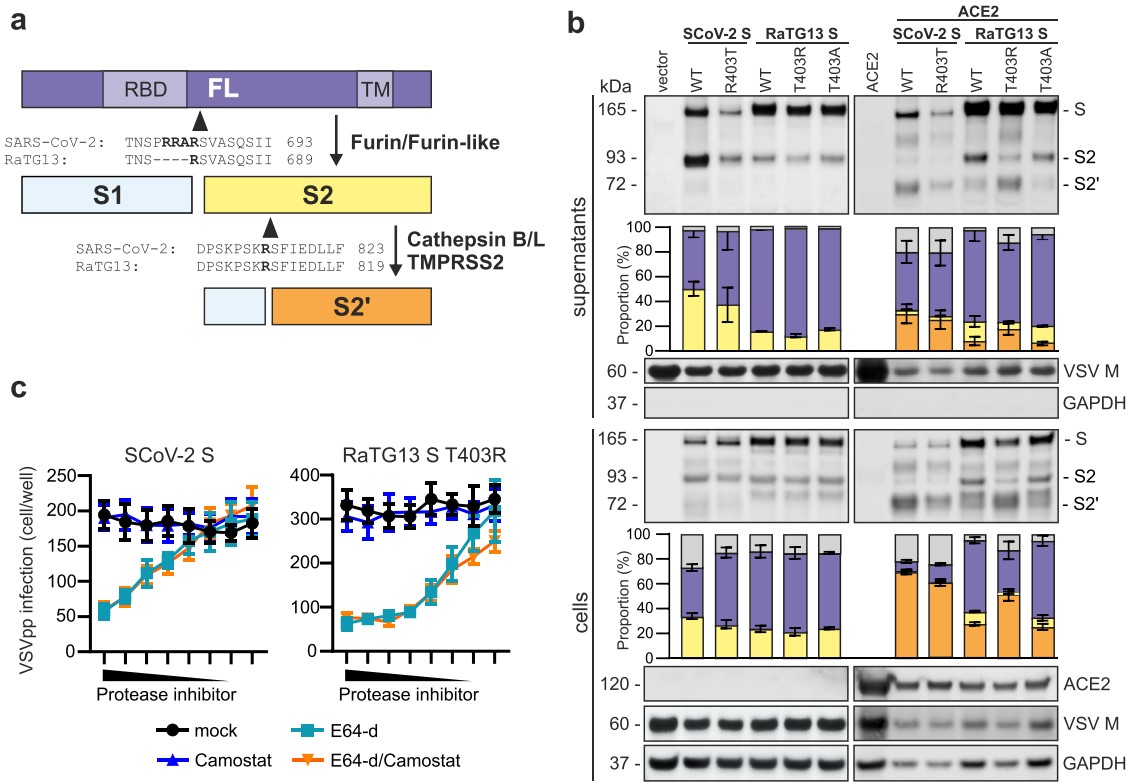

**Fig. 4 Interaction with ACE2 enhances S processing. a** Schematic representation of the SARS-CoV-2 S protein: FL (purple), S2 (yellow), S2′ (orange), and sequence alignments of the SCoV-2 and RaTG13 S1/S2 and S2′ cleavage sites. **b** Exemplary immunoblots of whole cells lysates (WCLs) and supernatants of HEK293T cells expressing SCoV-2 S, RaTG13 S, or the indicated mutant that were infected with VSVΔG-GFP in the absence (left) or the presence of a vector expressing human ACE2 (right). Blots were stained with anti-SCoV-2 S, anti-GAPDH, anti-ACE2, and anti-VSV-M. and quantified for Spike FL, S2, and S2′ expression. Bars represent the mean of three independent experiments (±SEM). **c** Automated quantification of GFP fluorescence of A549 ACE2 cells infected with VSVΔG-GFP pseudotyped with indicated SCoV-2 or RaTG13 S in the absence of ACE2. The cells were pre-treated (30 min) with 20 μM of E64-d or Camostat in the highest concentration and diluted in a 1:2 titrational row. Lines represent the mean of three independent experiments (±SEM).

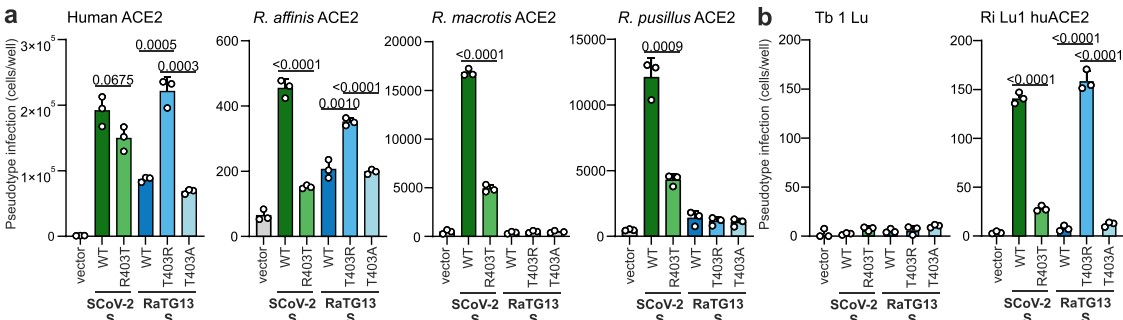

**Fig. 5 SCoV-2 S and T403R RaTG13 S allow entry with human but not bat ACE2. a** HEK293T cells expressing indicated ACE2 variants or **b** Tb 1 Lu, *Tadarida brasiliensis* derived lung epithelial and Ri 1 Lu huACE2 *Rhinolophus affinis* derived lung epithelial cells expressing human ACE2 were infected with VSVΔG-GFP pseudotyped with SCoV-2, RaTG13 or indicated mutant S. Quantification by automatic counting of GFP positive cells. Bars represent the mean of three independent experiments (±SEM). **a, b** Two-tailed Student's t-test with Welch's correction.

present study. It is tempting to speculate that these individuals might have a reduced risk for infection and/or severe COVID-19.

A recent study proposed that residue 501 plays a key role in the ability of RaTG13 S to use human ACE2 for viral entry[35] but the reported enhancing effect of changes at position 501 was weaker than that observed for the T403R change analyzed in the present study. However, the previous finding that numerous residues in the SARS-CoV-2 S RBD are involved in the interaction with the human ACE2 orthologue explains why the R403T substitution only moderately reduced SARS-CoV-2 infection. Mutation of E37 in ACE2 reduced the levels of WT CoV-2 S-mediated infection to

those obtained for the R403T CoV-2 S (Fig. 3a). This suggests that the interaction between R403 in the S protein and E37 in the ACE2 receptor is relevant for full infectiousness of SARS-CoV-2, although reduced expression levels and incorporation of the R403T mutant CoV-2 S may also contribute. More importantly, the reverse T403R substitution generally strongly enhanced RaTG13 S-mediated VSVpp infection without affecting protein expression levels.

SARS-CoV-2 entry requires sequential cleavage of the S protein at the S1/S2 and the S2′ cleavage sites to mediate membrane fusion and infection[13,20]. The RaTG13 S lacks the polybasic

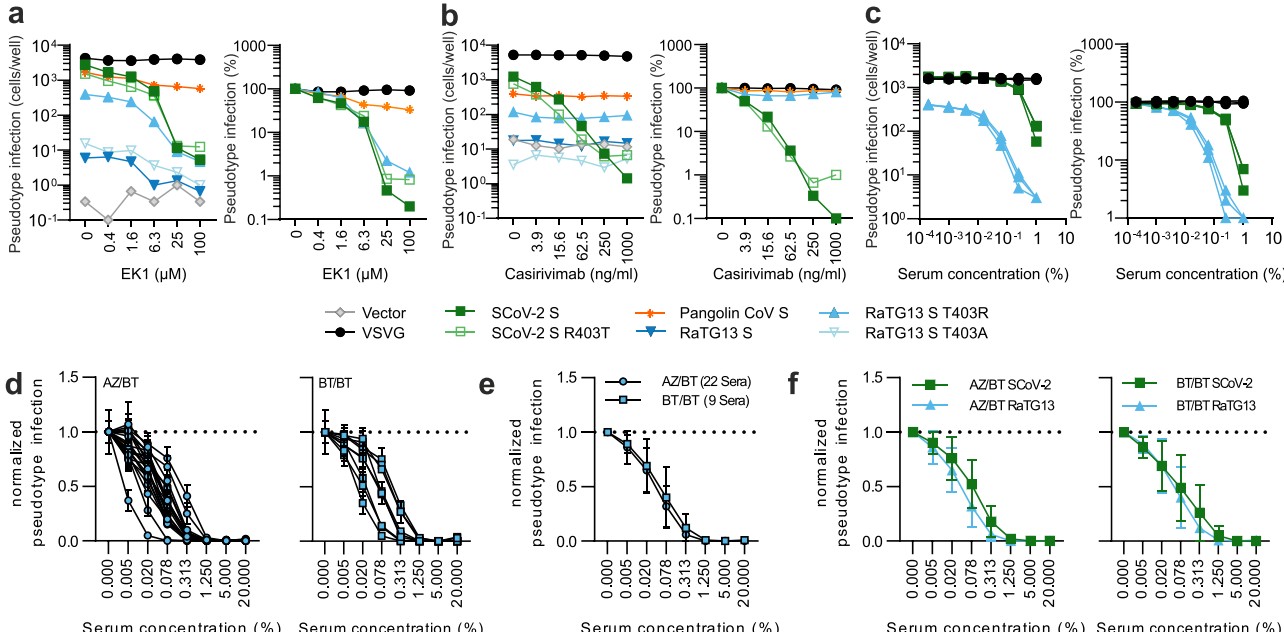

**Fig. 6 T403R RaTG13 S is sensitive to EK-1 and sera from vaccinated individuals but not Casivirimab. a−c** Automated quantification of GFP fluorescence of Caco-2 cells infected with VSVΔG-GFP pseudotyped with indicated S variants. The virus was pre-treated (30 min) with the indicated amounts of **a** EK-1 **b** Casivirimab (orange), and **c** Sera of BNT162b2 vaccinated patients. Lines represent the mean of three independent experiments (±SEM). **d** Sera of AZ/BNT162b2 and 2xBNT162b2 vaccinated patients with VSVΔG-GFP pseudotyped with RaTG13 S. Lines represent the mean of three biological replicates. Mean of the inhibition of VSVΔG-GFP pseudotyped with **e** RaTG13 S by Sera of AZ/BNT162b2 or 2x BNT162b2 vaccinated patients and **f** RaTG13 S or SCoV-2 by the Sera of AZ/BNT162b2 (left) or 2x BNT162b2 (right) vaccinated patients Lines represent the mean of three independent experiments (±SEM).

insertion at the S1/S2 site that enhances furin-mediated cleavage of the SARS-CoV-2 S (Fig. 4a). Thus, it was expected that the efficiency of proteolytic processing of RaTG13 S proteins at the S1/S2 site is reduced (Fig. 4b). It came as surprise, however, that overexpression of TMPRSS2 only enhanced VSVpp infection mediated by the SARS-CoV-2 but not by the RaTG13 T403R S protein (Fig. 2b) since the S2′cleavage site that is targeted by TMPRSS2 is identical in these S proteins. Instead, processing of the RaTG13 S at the S2′site was dependent on Cathepsins, which have also been shown to efficiently cleave the SARS-CoV-2 S in some cell types[27]. SARS-CoV-2 replicates in ACE2 expressing cells. In agreement with published data[21], the presence of ACE2 strongly enhanced proteolytic processing of the SARS-CoV-2 S (Fig. 4b). Similar effects were observed for RaTG13 S T403R but not on the WT RaTG13 S suggesting that interaction with ACE2 has a major impact on proteolytic processing.

It has been shown that the RBD of SARS-CoV-2 S shows higher homology to the corresponding region of the pangolin CoV S protein or the recently discovered BANAL-20-52 isolate from *Rhinolophus malayanus* in Laos than to RaTG13[5,6,36]. Whether or not this is a consequence of recombination or convergent evolution is under debate[37,38]. Notably, both BANAL-20-52 and Pan CoV S protein contain a positively charged residue at position 403 and are capable of utilizing human ACE2 for infection (Fig. 6a, b)[36]. It was suggested BANAL-20-52 is closer relative to SARS-CoV-2 than RaTG13, indicating that a change in residue 403 may have occurred before the emergence of this strain. Altogether our results suggest that a positively charged amino acid residue at position 403 in the S protein was a prerequisite for efficient zoonotic transmission and pandemic spread of SARS-CoV-2. Notably, a positively charged residue at the corresponding position is present in S proteins of most RaTG13-related bat coronaviruses (Supplementary Fig. 7). Thus, many bat sarbecoviruses, including the unknown precursor of SARS-CoV-

2, may be fitter for zoonotic transmission than RaTG13. However, our preliminary data suggest that vaccination might protect us against such close bat relatives of SARS-CoV-2.

## Methods

**Cell culture and viruses.** All cells were cultured at 37 °C in a 5% CO2 atmosphere. Human embryonic kidney 293T cells purchased from American type culture collection (ATCC: #CRL3216) were cultivated in Dulbecco's Modified Eagle Medium (DMEM, Gibco) supplemented with 10% (v/v) heat-inactivated fetal bovine serum (FBS, Gibco), 2 mM L-glutamine (PANBiotech), 100 µg/ml streptomycin (PAN-Biotech) and 100 U/ml penicillin (PANBiotech). Calu-3 (human epithelial lung adenocarcinoma, kindly provided and verified by Prof. Frick, Ulm University) cells were cultured in Minimum Essential Medium Eagle (MEM, Sigma) supplemented with 10% (v/v) FBS (Gibco) (during viral infection) or 20% (v/v) FBS (Gibco) (during all other times), 100 U/ml penicillin (PAN-Biotech), 100 µg/ml streptomycin (PAN-Biotech), 1 mM sodium pyruvate (Gibco), and 1 mM NEAA (Gibco). Caco-2 (human epithelial colorectal adenocarcinoma, kindly provided by Prof. Holger Barth, Ulm University) cells were cultivated in DMEM (Gibco) containing 10% FBS (Gibco), 2 mM glutamine (PANBiotech), 100 µg/ml streptomycin (PANBiotech), 100 U/ml penicillin (PANBiotech), 1 mM Non-essential amino acids (NEAA, Gibco), 1 mM sodium pyruvate (Gibco). A549, A549 ACE2, A549 TMPRSS2 and A549 ACE2/TMPRSS2 cells were cultured in DMEM (Gibco) supplemented with 10% (v/v) FBS (Gibco), 100 U/ml penicillin (PAN-Biotech), 100 µg/ml streptomycin (PAN-Biotech) and 10 µg/ml Puromycin for A549 ACE2 and A549 ACE2/TMPRSS2 and 10 µg/ml Blasticidin for A549 TMPRSS2 and A549 ACE2/TMPRSS2. I1-Hybridoma cells were purchased from ATCC (#CRL-2700) and cultured in RPMI supplemented with 10% (v/v) heat-inactivated FBS (Gibco), 2 mM L-glutamine (PANBiotech), 100 µg/ml streptomycin (PANBiotech), and 100 U/ml penicillin (PANBiotech). Tb 1 Lu (*Tadarida brasiliensis* derived lung epithelial) and Ri 1 Lu huACE2 (*Rhinolophus affinis* derived lung epithelial cells expressing human ACE2, kindly provided by Marcel A. Müller) were cultured in DMEM supplemented with 10% (v/v) heat-inactivated FBS (Gibco), 2 mM L-glutamine (PANBiotech), 100 µg/ml streptomycin (PANBiotech) and 100 U/ml penicillin (PANBiotech), 2 mM sodium pyruvate (Gibco). Viral isolates BetaCoV/France/IDF0372/2020 (#014V-03890) was obtained through the European Virus Archive global.

**Pseudoparticle production.** To produce pseudotyped VSVΔG-GFP particles, 6*10^6 HEK 293T cells were seeded 18 h before transfection in 10 cm dishes. The cells were transfected with 15 µg of a glycoprotein expressing vector using PEI (PEI,

1 mg/ml in H$_2$O, Sigma-Aldrich). 24 h post-transfection, the cells were infected with VSVΔG-GFP particles pseudotyped with VSV G at a MOI of 3. One hour post-infection, the inoculum was removed. Pseudotyped VSVΔG-GFP particles were harvested 16 h post-infection. Cell debris was pelleted and removed by centrifugation (500 × g, 4 °C, 5 min). Residual input particles carrying VSV-G were blocked by adding 10% (v/v) of I1 Hybridoma Supernatant (I1, mouse hybridoma supernatant from CRL-2700; ATCC) to the cell culture supernatant.

**Molecular dynamics simulation.** Based on the structure of ACE2-bound to SARS-CoV-2 spike taken from the Protein Data Bank[39]: 7KNB (https://www.rcsb.org/structure/7KNB), the initial atomic positions were obtained. Equilibration (300 K for 0.5 ns) was performed by ReaxFF (reactive molecular dynamic) simulations[40] within the Amsterdam Modelling Suite 2020 (http://www.scm.com). Based on the equilibrated structure, amino acids from the spike protein were replaced with the respective amino acids from RaTG13, respectively the modification. After an additional equilibration (300 K for 0.5 ns) ReaxFF (reactive molecular dynamic) simulations were performed within the *NVT* ensemble over 25 ps, while coupling the system to a Berendsen heat bath (*T* = 300 K with a coupling constant of 100 fs). The interaction energy was obtained by averaging over these simulations. For all visualizations the Visual Molecular Dynamics programme (VMD 1.9.3)[41] was used.

**Expression constructs.** pCG_SARS-CoV-2-Spike-IRES_eGFP, coding the spike protein of SARS-CoV-2 isolate Wuhan-Hu-1, NCBI reference Sequence YP_009724390.1, was provided by Stefan Pöhlmann (Göttingen, Germany). pCG_SARS-CoV-2-Spike C-V5-IRES_eGFP and RaTG13-S (synthesized by Baseclear) were PCR amplified and subcloned into a pCG-IRES_eGFP expression construct using the restriction enzymes XbaI and MluI (New England Biolabs). The SARS-CoV-2 S R403T and RaTG13 S T403R/T403A mutant plasmids were generated using Q5 Site-Directed Mutagenesis Kit (NEB). ACE2 was synthesized by Twist bioscience, PCR amplified, and subcloned into a pCG- expression construct using the restriction enzymes XbaI and MluI (New England Biolabs). ACE2 E37A, ACE2 D38A, and ACE2 D37A/D38A mutations were introduced by Splice-varlap Extention PCR using the Thermo scientific Phusion High-Fidelity PCR Kit and subcloned into a pCG-expression construct. All constructs were verified by sequence analysis. Primer sequences are listed in Supplementary Data 2.

**Transfections.** Plasmid DNA was transfected using either calcium phosphate transfection or Polyethylenimine (PEI, 1 mg/mL in H$_2$O, Sigma-Aldrich) according to the manufacturer's recommendations or as described previously[42].

**Biosafety.** All work with bacmid-generated SARS-CoV-2 virus was performed in the registered BSL3 facility at the Institute of Virology at University Hospital of the Friedrich-Alexander-University Erlangen-Nuremberg (Az. 821-8760.00-23/90; Az. 821-8791.2.12; Az. 821-8791.2.13). Generation of recombinant SARS-CoV-2 by the bacmid method was approved by the Central Committee for Biological Safety (ZKBS) consulting the German Federal Office of Consumer Protection and Food Safety (BVL) (Az. 45110.2084). Permission for the experiments was granted by the regional authorities of Lower Franconia (Az. 55.1-8791.27-28-20 and Az. 55.1-8791.27-29-20).

**Cloning of SARS-CoV-2 ΔS bacmid.** An anonymized residual respiratory swab sample from a patient with SARS-CoV-2 infection was used as a template for genome amplification. Total nucleic acids were extracted on an automated Qiagen EZ1 robotic workstation using the Qiagen EZ1 virus mini kit v2.0 according to the manufacturer's instructions. Genomic viral RNA was reverse transcribed using the NEB LunaScript RT SuperMix Kit according to the manufacturer's protocol. Four overlapping fragments covering the entire viral genome were amplified using the NEB Q5 High-Fidelity DNA Polymerase. The resulting amplicons were assembled with a modified pBeloBAC11 backbone, containing CMV and T7 promotors as well as the HDV ribozyme and bGH polyA signal, using the NEBuilder HiFi DNA Assembly Cloning Kit. Assembled DNA was electroporated into *E. coli* GS1783 strain and resulting clones of pBelo-SARS-CoV-2 were confirmed by restriction digestion and next-generation sequencing. The viral Spike gene was replaced with a kanamycin-cassette flanked by SacII restriction sites by homologous recombination using the Lambda-Red Recombination System[43]. The bacmid was linearized with the restriction enzyme SacII, and EGFP or GLuc reporter cassettes were introduced instead of Spike using the NEBuilder HiFi DNA Assembly Cloning Kit according to the manufacturer's instruction. Positive clones were confirmed by restriction digestion and sequencing.

**SARS-CoV-2 ΔS replicon system.** HEK293 T cells were seeded in six-well format and transfected with 3 μg pBelo-SARSCoV-2-dSpike-GLuc-K2 or pBelo-SARS-CoV-2-dSpike-EGFP and 0.25 μg of each expression construct pLVX-EF1alpha-SARS-CoV2-N-2xStrep-IRES-Puro, pCG-ACE2, pCAG-T7-RNA-polymerase, and one pCG- vector encoding the spike protein of SARS-CoV-2, RaTG13 or the indicated mutant S respectively. Two days after transfection, bright field and fluorescence microscopy (GFP) images were acquired using the Cytation 3 microplate reader (BioTek). Gaussia luciferase activity in the supernatants was

measured with the Gaussia Luciferase Assay system (Promega) according to the company's instructions.

**Endpoint titration to determine viral titers.** To determine viral titers for subsequent equal infections, viral stocks were serial diluted and Caco-2 cells were infected with different dilutions. Two days post infection cells were fixed with 4% PFA (Sigma-Aldrich), washed with PBS-T, and analyzed using the ELISpot/Immunospot® S6 ULTIMATE UV Image Analyzer (Cellular Technology Limited/CTL, USA). The dilution where half of the wells are still infected was used for the calculation of stock concentration.

**Cloning and reconstitution of rec-SARS-CoV-2 d6-YFP and d6-YFP-R403T.** As previously described[18], an anonymized residual sample from a patient with SARS-CoV-2 infection was used as a template for genome amplification. Total nucleic acids of a respiratory swab sample were extracted on an automated Qiagen EZ1 analyzer using the Qiagen EZ1 virus mini kit v2.0 according to the manufacturer's instructions. Genomic viral RNA was reverse transcribed using the LunaScript RT SuperMix Kit (NEB) according to the manufacturer's protocol. Four overlapping fragments covering the entire viral genome were amplified using the Q5 High-Fidelity DNA Polymerase (NEB). The resulting amplicons were assembled with a modified pBeloBAC11 backbone (NEB, GenBank Accession #: U51113) containing CMV and T7 promotors and bGH polyA signal (amplified from pcDNA4, Thermo Fisher Scientific, V86320) as well as the HDV ribozyme (synthesized as gene block by Integrated DNA Technologies IDT) using the NEBuilder HiFi DNA Assembly Cloning Kit (NEB). Assembled DNA was electroporated into the E. coli GS1783 strain and resulting clones of pBSCoV2 were confirmed by restriction digestion and next-generation sequencing (MiSeq™, Illumina).

The viral ORF6 gene was replaced with EYFP by homologous recombination using the two-step Lambda-Red Recombination System (Tischer et al., 2010; PMID: 20677001), resulting in pBSCoV2-d6-YFP. The spike R403T point mutation was generated by two-step Lambda-Red Recombination using pBSCoV2_d6-YFP as template, resulting in pBSCoV2_d6-YFP_Spike R403T. Restriction digestion and next-generation sequencing confirmed positive clones of both bacmids.

For virus reconstitution, a co-culture of HEK293T cells stably expressing either ACE2 (Hoffmann, 2020; PMID: 32142651; cloned into pLV-EF1a-IRES-Blast, addgene #85133) or the viral N protein (amplified from patient material; cloned into pLV-EF1a-Blast) and T7-RNA polymerase (amplified from pCAGT7, kind gift from Marco Thomas; cloned into pLV-EF1a-IRES-Puro, addgene #85132) was transfected with pBelo-S-CoV-2-d6-YFP or d6-YFP_Spike R403T using GenJet™ Reagent (II) (SL100489, SignaGen® Laboratories, Frederick, MD, USA) according to the manufacturer's protocol. Three days post-transfection, the supernatant was transferred on CaCo-2 cells for passage 1 (P1) virus stocks. Passage 2 (P2) virus stocks were obtained after infection of CaCo-2 cells with 1:50 volume of P1 virus. Viral titers were determined by endpoint titration on CaCo-2 cells.

**Replication kinetics.** To analyze growth kinetics, Caco-2 cells were infected with either SARS-CoV-2 d6-YFP wild type, SARS-CoV-2 d6-YFP R403T, or mock at an MOI of 0.005 or 0.02. Therefore, cells were seeded in 96-well format with a cell density of 2,5 × 10$^4$/100 μl and 24 h after seeding infected with respective MOIs. Supernatant samples were taken at 0, 10, 24, 34, 48, and 72 h post infection and diluted 1:10 with H$_2$O. Samples were digested with proteinase K (Roche, Switzerland) for 1 h at 56 °C, and proteinase was heat-inactivated. Afterwards, digested samples were utilized for RT-qPCR using the Universal Probe One-Step RT-qPCR kit (NEB, USA) according to manufacturer's instructions in an Applied Biosystems 7500 Real Time PCR system (Applied Biosystems, USA). Probes targeting viral polymerase (RdRP) were 5′-labelled with VIC (2′-chloro-7'phenyl-1,4-dichloro-6-carboxy-fluorescein) and 3′-labelled with BMN535 quencher modification (Biomers, Ulm) (Primers used: fwd: 5′-gtgaaatggtcatgtgtggccg-3', rev: 5′-caaatgttaaaaacactattagcata-3; probe: 5′-caggtggaacctcatcag-gagatgc3')

**Whole-cell and cell-free lysates.** Whole-cell lysates were prepared by collecting cells in phosphate-buffered Saline (PBS, Gibco), pelleting (500 g, 4 °C, 5 min), lysing, and clearing as previously described[42]. Total protein concentration of the cleared lysates was measured using the Pierce BCA Protein Assay Kit (Thermo Scientific) according to the manufacturer's instructions. Viral particles were filtered through a 0.45 μm MF-Millipore Filter (Millex) and centrifuged through a 20% sucrose (Sigma) cushion. The pellet was lysed in transmembrane lysis buffer already substituted with Protein Sample Loading Buffer (LI-COR).

**SDS-PAGE and immunoblotting.** SDS-PAGE and immunoblotting was performed as previously described[42]. In brief, whole-cell lysates were separated on NuPAGE 4-12% Bis-Tris Gels (Invitrogen) for 90 min at 120 V and blotted at constant 30 V for 30 min onto Immobilon-FL PVDF membrane (Merck Millipore). After the transfer, the membrane was blocked in 1% Casein in PBS (Thermo Scientific) and stained using primary antibodies directed against SARS-CoV-2 S (1:1,000, Biozol, 1A9, #GTX632604), α-SARS-CoV-2 N Sino Biologicals #40588-V08B (1:1000), anti-SCoV 2 nucleocapsid Invitrogen, #ARC2372 (1:2000), ACE2 (1:1,000, Abcam, #GTX632604), VSV-M (1:2,000, Absolute Antibody, 23H12, #Ab01404-2.0), V5-

tag (1:1,000, Cell Signalling, #13202), Anti-HSP70 Santa Cruz W27/sc-24 (1:1000), Anti-GFP GenScript A01704-40 (1:1000), GAPDH (1:1,000, BioLegend, #631401), secondary Anti-human-488 Invitrogen A11013 (1:2000), Anti-mouse-647 Invitrogen A31571 (1:2000), Anti-rabbit-647 Invitrogen A21206 (1:2000) and Infrared Dye labelled secondary antibodies IRDye 800CW Goat anti-Mouse #926-32210, IRDye 800CW Goat anti-Rat (#926-32219), IRDye 680CW Goat anti-Rabbit (#925-68071), IRDye 680CW Goat anti-Mouse (#926-68070), IRDye 800CW Goat anti-Rabbit (#926-32211) all 1:10,000. Proteins were detected using a LI-COR Odyssey scanner and band intensities were quantified using LI-COR Image Studio version 5.

**ACE2 interaction assay.** HEK293T cells expressing the indicated S constructs were collected after 48 h in phosphate-buffered saline (PBS, Gibco), divided for simultaneous immunoblotting and ACE2 interaction assay, and pelleted ($500 \times g$, 4 °C, 5 min). For the ACE2 interaction assay, the cells were lysed in NP-40 buffer (150 mM NaCl, 1% (vol/vol) NP-40, 50 mM HEPES pH 7.4 and protease inhibitor cocktail (Sigma)) and cell debris pelleted by centrifugation at $20,000 \times g$ for 20 min at 4 °C. ACE2 coated wells (COVID-19 Spike-ACE2 binding assay II, Ray Bio) were incubated for 2 h with 50 µl of WCLs and washed extensively with the provided wash buffer (RayBio, #EL-ITEMB). The wells were incubated 1 h with 100 µl anti-V5(MS) (1:1,000, Cell Signalling, #80076), subsequently washed and incubated for 1 h with 100 µl anti-MS-HRP (1:1,000, RayBio). After washing, the samples were incubated with 50 µl of TMB Substrate Solution (RayBio, #EL-TMB) for 30 min. The reaction was stopped by adding 50 µl Stop Solution (RayBio, #EL-STOP) and absorption was measured at 450 nm with a baseline correction at 650 nm.

**Stem cell culture and intestinal differentiation.** Human embryonic stem cell line HUES8 (Harvard University, Cambridge, MA) was used with permission from the Robert Koch Institute according to the "79. Genehmigung nach dem Stammzellgesetz, AZ 3.04.02/0084". Cells were cultured on human embryonic stem cell matrigel (Corning, Corning, NY) in mTeSR Plus medium (STEMCELL Technologies, Vancouver, Canada) at 5% CO2, 5% O2, and 37 °C. Medium was changed every other day and cells were split with TrypLE Express (Invitrogen, Carlsbad, CA) twice a week. For differentiation, 300,000 cells per well were seeded in 24-well plates coated with growth factor-reduced matrigel (Corning) in mTeSR Plus with 10 mM Y-27632 (STEMCELL Technologies). The next day, differentiation was started at 80−90% confluency, as described previously[44].

**Intestinal organoids.** To prepare in vitro differentiated organoids for transduction, matrigel was dissolved in Collagenase/Dispase (Roche, Basel, Switzerland) for 2 h at 37 °C and stopped by cold neutralization solution (DMEM, 1% bovine serum albumin, and 1% penicillin-streptomycin). Organoids were transferred into 1.5 mL tubes and infected in 300 µl pseudoparticle containing inoculum. Organoids were then resuspended in 35 µL cold growth factor–reduced matrigel to generate cell-matrigel domes in 48-well plates. After 10 min at 37 °C, intestinal growth medium (DMEM F12 [Gibco, Gaithersburg, MD], 1× B27 supplement [Thermo Fisher Scientific], 2 mM L-glutamine, 1% penicillin-streptomycin, 40 mM HEPES [Sigma-Aldrich], 3 µM CHIR99021, 200 nM LDN-193189 [Sigma-Aldrich], 100 ng/mL hEGF [Novoprotein, Summit, NJ], and 10 µM Y-27632 [STEMCELL Technologies]) was added and organoids were incubated at 37 °C. The organoids were imaged using the Cytation 3 cell imaging system and processed with Gen5 and ImageJ(Fiji) software. For FACS preparation, the matrigel was dissolved and the extracted organoids were dissolved in Accutase (Stemcell technologies). The cells were fixed with PBS for 10 min at 4 °C and washed with cold PBS containing 2% FBS. Flow cytometry analyses were performed using a FACS CANTO II (BD) flow cytometer. Transduction rates were determined by GFP expression and analyzed with BD FACSDiva and Flowjow10 software.

**Pseudoparticle inhibition.** VSVpp inhibition experiments were performed as previously described[45]. In brief, Caco-2 cells were infected with 100 µl freshly produced VSVΔG-GFP pseudo particles, which were preincubated (30 min, 37 °C) with the indicated amounts of Casivirimab (REGN10933, orange), EK-1 (Core Facility Functional Peptidomics (UPEP), Ulm University) or Sera of with BNT162b2 vaccinated donors. GFP-positive cells were automatically quantified using a Cytation 3 microplate reader (BioTek).

**Sera from vaccinated individuals.** Blood samples of ChAdOx1-nCoV-19/BNT162b2 and BNT162b2 vaccinated individuals were obtained after the participants information and written consent. Samples were collected 13−30 days after the second dose using S-Monovette Serum Gel tubes (Sarstedt). Before use, the serum was heat-treated at (56 °C, 30 min). Ethics approval was given by the Ethic Committee of Ulm University (vote 99/21− FSt/Sta).

**Treatment with ATN-161.** Caco-2 cells were preincubated with the indicated amounts of α5β5 integrin Inhibitor ATN-161 (Sigma) for 2 h and infected with 100 µl freshly produced VSVΔG-GFP pseudo particles. Sixteen h post-infection, GFP positive cells were automatically quantification using a Cytation 3 microplate reader (BioTek). Calu-3 cells were preincubated with the indicated amounts of ATN-161 (Sigma) for 2 h and infected with SARS-CoV-2 isolate BetaCoV/France/IDF0372/2020 (MOI 0.05, 6 h). Forty-eight hours post-infection supernatants were harvested for qRT-PCR analysis.

**Sequence logo and alignments.** Alignments of primary coronavirus sequences: https://www.ncbi.nlm.nih.gov/nuccore/(GU190215.1/QPD89842.1/KY417145.1/KY938558.1/KU182964.1/KJ473811.1/MN996532.1/KY417152.1/MK211376.1/KY417146.1/KY417150.1/KT444582.1/MG772933.1/MG772934.1/KY770858.1/KY770859.1/KJ473816.1/MK211374.1/JX993987.1/KJ473814.1/DQ648856.1/KY770860.1/KJ473812.1/KJ473813.1/JX993988.1/KY417143.1/MK211378.1/DQ412043.1/DQ648857.1/KY417148.1/MK211375.1/KY417147.1/KY417142.1/MK211377.1/KJ473815.1/GQ153542.1/GQ153543.1/KF294457.1/GQ153547.1/DQ084199.1/GQ153548.1/DQ022305.1/DQ084200.1/GQ153545.1/GQ153546.1/GQ153539.1/GQ153540.1/GQ153541.1/GQ153544.1) were performed using ClustalW[46] with a gapOpening penalty of 80. Sequence logos were generated using R packages ggplot2 and ggseqlogo[47].

**Statistics.** Statistical analyses were performed using GraphPad PRISM 9.2 (GraphPad Software). *P*-values were determined using a two-tailed Student's t-test with Welch's correction. Unless otherwise stated, data are shown as the mean of at least three independent experiments ± SEM. See supplemental Data 1 for detail.

**Reporting summary.** Further information on research design is available in the Nature Research Reporting Summary linked to this article.

## Data availability

The datasets generated during and/or analyzed during the current study are available from the corresponding authors on request. All Source data are provided in the Supplementary Information/Source Data file. The SARS-CoV-2 S structure used in this study is available in the Protein Data Bank (PDB) under accession code 7KNB. Source data are provided with this paper.

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

## Acknowledgements

We thank Kerstin Regensburger, Regina Burger, Jana-Romana Fischer, Birgit Ott, Martha Meyer, Nicole Schrott and Daniela Krnavek for technical assistance. The ACE2 vector, SARS-CoV-2 S-HA plasmid, and A459 cells were kindly provided by Shinji Makino and Stefan Pöhlmann, and bat cells by Marcel A. Müller. F.Z., C.P.B., J.K., and L.K. are part of the International Graduate school for Molecular Medicine (IGradU), Ulm. This study was supported by DFG grants to F.K. (CRC 1279, SPP 1923), K-K.C. (Co260/6-1 Neuro-COVID), T.J. (CRC1279), A.K. (KL 2544/8-1, KL 2544/5-1,7-1 and the 'Heisenberg-Programm' KL 2544/6-1), E.S-G (CRC1279, EXC 2033-390677874 and 436586093) and K.M.J.S. (CRC1279, SPP1923, SP1600/6-1). A.E. is funded by the DFG (EN-423/7-1) as well the State of Bavaria "BAY-VoC" and "Coronaforschung". F.K., K.M.J.S., and A.E. were supported by the BMBF (Restrict SARS-CoV-2, IMMUNOMOD, and 01KI20172A SENSE-CoV2).

## Author contributions

F.Z. performed most experiments with support by R.N. and C.B.P. D.S., M.V., Q.X., and L.K. performed western blots and interaction assays. A.H. and A.C. generated and analyzed SARS-CoV-2 bacmids and recombinant viruses. J.K., S.H., and A.K. contributed the gut organoids. S.P. and M.H. generated the A459-ACE2 cell line expressing TMPRSS2 and provided reagents. C.J. and T.J. performed molecular modelling analyses. K.-K.C. provided VSV pseudotypes and reagents. S.S. provided resources and J.M. serum samples. F.Z., D.S., K.M.J.S., A.E. and F.K. conceived the study, planned experiments, and wrote the manuscript. All authors reviewed and approved the manuscript.

## Funding

## Competing interests

The authors declare no competing interests.
