## [Peer Review File · Nature Communications]

Spike residue 403 affects binding of coronavirus spikes to human ACE2Reviewers' Comments:

Reviewer #1:

Remarks to the Author:

SARS-CoV-2 is believed originated from bats and spread to human through intermediate host. Mutations in S protein achieve virus cross-species transmission. In this article, Fabian et al., reported the T403R mutation strengthens the ability of the RaTG13 S protein to bind human ACE2 and enhances infection of RaTG13 S to human cells and intestinal organoids, which provides insight for SARS-CoV-2 cross-species transmission. But a lot of questions need to be addressed to support the conclusion.

1. The key residues determine the binding affinity between SARS-CoV-2-S and ACE2 were previously reported (Lan et al., Nature. 2020 May;581(7807):215-220. Shang et al., Nature. 2020 May;581(7807):221-224.) and the variants of concern currently contains the mutations of key residues for virus infection, as well as for receptor binding, like 501Y, 484K, 478K, 417N. Why the authors only choose R403? What's the contribute of other key residues to RaTG13 S binding to hACE2?
2. The authors are suggested to do biochemical assays to show the binding affinities between S or S-403 variants and ACE2.
3. For Fig.S3, the WCLs of different cells didn't treated by additional proteases. What protease cleaved the S into S2'?
4. An unrelated protein (like DPP4) should be co-expressed with S as a negative control to ACE2/S coexpression.
5. The authors are suggested to do proteolytic assay with purified VSVpp and ACE2 to confirm the conclusion T403R promotes RaTG13 S processing in the presence of human ACE2.
6. More experiments are need to support the conclusion T403R allows RaTG13 S to spread to different human organs. T403R mutation increased the binding affinity of hACE2, which could allow RaTG13 to infect intestinal organoids derived from pluripotent stem cells. This could only indicate T403R change increase RaTG13 infection but not spread to different organs.
7. Could the author explain why the sera from SARS-COV-2 S vaccinated individuals have 10-fold higher neutralization efficiency to T403R RaTG13 S than SARS-COV-2 S? There's only one point mutation on RaTG13 S. Could the authors compare the binding affinity between S or S variants and antibodies?
8. The labels for S2 and S2' bands in Fig.S3 are upside down.
9. For Fig.S5 b, please check the labels for RaTG13 and its mutations, it's not corresponding to Fig. S5 a. Also, there's Fig.S5 c in the figure legend but no figure.

Reviewer #2:

Remarks to the Author:

In the manuscript entitled "Spike mutation T403R allows bat coronavirus RaTG13 to use human ACE2" Zech and colleagues identified that the residue at position 403 in the spike protein of SARS-CoV-2 and the related bat-derived RaTG13 coronavirus plays an important role in regulating binding to the human ACE2 receptor.

Using reactive force field simulations authors showed that when an arginine (R) is present at position 403, as observed for SARS-CoV-2 spike, the spike-ACE2 interface is stabilized by charge interactions whereas when a threonine (T) is present, as is the case for RaTG13 spike, the interaction appears destabilized.

Infection of Caco-2 cells by VSV-based pseudotyped particles bearing wild-type (wt) or mutated SARS-CoV-2 spike protein showed that wild-type R403 favored infectivity while the R403T mutation decreased infection. Conversely, wt RaTG13 spike pseudotyped particles bearing T403 were associated with low infectivity, however the T403R mutation allowed for a substantial enhancement of infectivity while T403A mutation did not result in any increase. Similar results were obtained using a complementation assay based on a SARS-CoV-2 Δ S replicon system in HEK-293T cells expressing

ACE2. The T403R mutation was also found to increase RaTG13 spike proteolytic processing in presence of human ACE2.

Infectivity findings using the Caco-2 cell line were further confirmed by infection of human gut organoids with VSV-based pseudotyped particles bearing SARS-CoV-2 and RaTG13 spike proteins. Using the VSV-based pseudotyping system, authors also tested ACE2 receptor usage from different bat species showing modest infectivity of wt SARS-CoV-2 spike pseudotyped particles in transfected cells expressing ACE2 from bats of the genus *Rhinolophus*. The T403R mutated RaTG13 spike was also found to be sensitive to EK-1 fusion inhibitor treatment as well as from sera of vaccinated individuals.

Overall, this is a well-conceived and executed study which brings to light new insights into the molecular determinants of ACE2 receptor binding for SARS-CoV-2 and the related bat coronavirus RaTG13. There are several key points and figures that would require some clarification in particular the choice of cell types and drugs used in the study along with how some of the data are interpreted. Detailed comments and suggestions for improvements are found below.

Main comments

1. Throughout most of this work, intestinal Caco-2, HEK-293T, and gut organoids were used for infectivity assays with the exception of lung-derived Calu-3 cells which were used in supplementary fig. 4. While it is well-established that SARS-CoV-2 can productively infect enterocytes and cell types other than lung cells, can authors replicate findings obtained from Caco-2 and gut organoids in lung-derived cells, in particular infectivity assays? As coronavirus route of entry is often cell-type dependent, the choice of cell types used here should at least be discussed.

2. For fig. 4a, the titles "Pu ACE2" for *Rhinolophus affinis* ACE2 "Ma ACE2" and "Rh ACE2" are confusing—consider clarifying. Overall for fig. 4 it may be best to separate panels into two figures with panels a and b (receptor usage) forming one figure and c, d, and e (VSV pseudoparticle inhibition assays) forming another one as the relationship between these two sets of data are not explained clearly in the manuscript. Indeed, the title for fig. 4 only mentions human/bat ACE2 usage and the manuscript text describes fig. 4 in two separate paragraphs. In addition, currently the title for fig. 4—line 491 "Fig. 4: SARS-CoV-2 S and T403R RaTG13 S allow entry with human but not bat ACE2." does not match what the data shown and with the statement in the manuscript text—lines 121-123: "Both WT SARS-CoV-2 S and (to a lesser extent) R403T SARS-CoV-2 S proteins were also capable of using bat (*Rhinolophus affinis*) ACE2 for viral entry although the overall infection rates were low (Fig. 4a). The figure title could mention that *Rhinolophus affinis* ACE2 allows some degree of infection that is lower than human ACE2.

3. EK-1 is a peptide-based fusion inhibitor that targets the heptad repeat 1 (HR1) region within spike trimers. HR1 is located in the S2 fusion domain while position 403 is in the RBD region of the S1 domain, so the functional link between the two is unclear. This point is not discussed in the manuscript; however this is important to interpret results appropriately. Could the authors provide more explanations as to the reasons why they chose this HR-targeting fusion inhibitor for testing the sensitivity of spike mutants at the distally-located position 403?

4. Supplementary fig. 2b clearly shows that the R403T mutant SARS-CoV-2 spike is incorporated much less efficiently into VSV particles than the wt spike. This begs the question of whether the 40% decrease in infectivity of SARS-CoV-2 R403T spike VSVpp compared to wt spike shown in fig. 2 is attributable to a modification in spike-ACE2 binding interface or simply because less mutated spike was incorporated into particles in the first place. This should be explained and discussed more explicitly in the manuscript.

5. In the discussion, authors state—lines 163-165: "Notably, a positively charged residue at the corresponding position is present in the S proteins of the great majority of RaTG13-related bat coronaviruses (Supplementary Fig. 6)". However, the data shown in supplementary fig. 6 for bat coronavirus sequences is confusing. In panel a, the sequence logo does not show "conservation of the

RGD motif in bat coronavirus spike proteins" as stated in the figure title. The red box shows that it's RGG that appears to be the most conserved motif. In panel b for bat coronavirus sequences the 3 residues in bold appears to be shifted one residue downstream and only 3 out of 9 sequences contain a basic residue that is stated in the text (R or K for BtCoV/Rp3, Bat SARS-like CoV, and Rhinolophus bat CoV) so not really a "a great majority". In addition to the improvement suggested above, this supplemental figure would benefit from substantial revisions, for instance by using a uniform naming system for all sequences used in panel b (SARS-CoV-2 strains are named by accession numbers while the other sequences are names of viral strains or isolates) as well as residue numbers at least for the start and the end of the truncated sequences used in the analysis. Also, what is the rationale behind the choice of 9 bat coronavirus sequences in panel b? Do all these bat coronaviruses belong to Sarbecovirus subgenus, what is their phylogenetic relationship with RaTG13 and SARS-CoV-2? It is unclear why panel c was added to supplementary fig. 6.

Minor comments

1. Lines 63-65: "We found that R403 is highly conserved in SARS-CoV-2 S proteins: only 233 of 1.7 million sequence records contain a conservative change of R403K and just 18 another residue or deletion." Was the R403T mutation found among the 18 other residue changes identified? If so, it would be worth mentioning or discussing.

2. In fig. 2c, the representative brightfield images shown do not allow to see much difference between conditions, especially the reduced CPE for SARS-CoV-2 R403T compared to wt SARS-CoV-2. This figure panel would benefit from being presented more clearly.

3. Some typos/mistakes identified:

- Line 118: "bat derived" the suggestion here is to add a hyphen: "bat-derived"
- Line 139: "SARS-Co-2" should be changed to "SARS-CoV-2"
- Line 161: "positive residue" should be changed to "positively charged residue"
- Line 203: "ACE2-bounded to SARS-CoV-2" should be changed to "ACE2 bound to SARS-CoV-2 spike"
- Line 221: "H2O" the 2 should be lower case
- Lines 232-233: "pBelo-SARSARS-CoV-2" ?
- Line 468: The indicated arrows in fig. 1 appears grey not "purple".

Reply to the reviewer's comments (in *italic* letters)

Reviewer #1: SARS-CoV-2 is believed originated from bats and spread to human through intermediate host. Mutations in S protein achieve virus cross-species transmission. In this article, Fabian et al., reported the T403R mutation strengthens the ability of the RaTG13 S protein to bind human ACE2 and enhances infection of RaTG13 S to human cells and intestinal organoids, which provides insight for SARS-CoV-2 cross-species transmission. But a lot of questions need to be addressed to support the conclusion.

1. The key residues determine the binding affinity between SARS-CoV-2-S and ACE2 were previously reported (Lan et al., Nature. 2020 May;581(7807):215-220. Shang et al., Nature. 2020 May;581(7807):221-224.) and the variants of concern currently contains the mutations of key residues for virus infection, as well as for receptor binding, like 501Y, 484K, 478K, 417N. Why the authors only choose R403? What's the contribute of other key residues to RaTG13 S binding to hACE2?

The S proteins of SARS-CoV-2, SARS-CoV-1 and Pangolin CoV all efficiently bind ACE2. Most variations between the RaTG13 and SARS-CoV-2 S receptor binding domains (RBDs) are present in either the SARS-CoV-1 or Pangolin CoV RBD. The exception in the RBD was R403T making it a strong candidate for the poor ability of the RaTG13 S to use human ACE2. As mentioned in the manuscript (lines 62-64), Cryo-EM structures and computational modelling further suggested that R403 might be involved ACE2 binding. We do not question the potential contribution of other, partially conservative aa exchanges in the SARS-CoV-2 RBD to human ACE2 binding. In fact, it has been reported that the combination of F449Y, L486F, Y493Q, Y498Q, D501N, and H505Y increased pseudoparticle infection efficiency in HELA-ACE2 cells by ~2-fold (Liu et al., 2021). However, our goal was to identify the reason for the poor activity of the RaTG13 S and we pinpointed it to T403.

2. The authors are suggested to do biochemical assays to show the binding affinities between S or S-403 variants and ACE2.

To address this, we established an in vitro ELISA assay. Our results show that the T403R change increases binding of RaTG13 S to the human ACE2 receptor (new Fig. 3c-e).

3. For Fig.S3, the WCLs of different cells didn't treated by additional proteases. What protease cleaved the S into S2'?

The impact of the T403R change on proteolytic activation of RaTG13 S is interesting and has been addressed by additional experiments and textual changes (lines 133-143; 151-163; 228-240). The S precursor is cleaved into S1 and S2 inside the cell by furin (e.g. Xia et al., Signal transduction and targeted therapy, 2020). Cleavage of RaTG13 S is less efficient because it lacks the polybasic site shown to be important for effective processing of the SARS-CoV-2 S (new Fig. 4a). The SARS-CoV-2 S2 can be cleaved to S2' to liberate the fusion peptide by TMPRSS2 during the entry process (lines 133-135). We now show that TMPRSS2 enhances infection mediated by the SARS-CoV-2 S but not by the RaTG13 S (new Fig. 2c) although the primary cleavage site is conserved (new Fig. 4a). It has been reported that cathepsins may compensate for TMPRSS2 (lines 152/153). Inhibition assays support that cathepsins activates the RaTG13 T403R S protein (new Figs. 4c; supplementary figure 6).

4. An unrelated protein (like DPP4) should be co-expressed with S as a negative control to ACE2/S coexpression.

We considered co-expression of an unrelated but decided against it because the sense of such a "control" is not clear to us. As is common practice, we always included the vector control for comparison. In addition, empty vector was used to adjust the total amount of transfected DNA to the same quantity in each transfection. Notably, it has been reported that DPP4 play a role in COVID-19 (Yang et al., PLOS One 2021; Strollo and Pozill, Diabetes Metab Res Rev. 2020).

5. The authors are suggested to do proteolytic assay with purified VSVpp and ACE2 to confirm the conclusion T403R promotes RaTG13 S processing in the presence of human ACE2.

We performed the suggested experiment with purified VSVpp. However, the levels of particles associate S protein were too low for meaningful analyses in this setting. To further address this point, we produced VSVpp

in the presence and absence of ACE2 and show that ACE2 strongly enhances processing of S2 to S2' in both the cells and the pseudoparticle containing supernatants (new Fig. 4b).

6. More experiments are need to support the conclusion T403R allows RaTG13 S to spread to different human organs. T403R mutation increased the binding affinity of hACE2, which could allow RaTG13 to infect intestinal organoids derived from pluripotent stem cells. This could only indicate T403R change increase RaTG13 infection but not spread to different organs.

We did not draw this conclusion but just stated that we assessed the possibility that the T403R change might allow the bat CoV RaTG13 to spread to different human organs by infection of organoids (lines 87-89). Enhanced infection should be associated with an increased potential to spread. However, the RaTG13 S VSVpp only allow to examine single-round infection. To substantiate the impact of the T403R changes, we now show that it strongly enhances infection of the human lung cancer cell lines Calu-3 and A549 (new Figs. 2b, 2c and Extended Data Fig. 1). We also show that substitution of R403T attenuates recombinant SARS-CoV-2 replication (new Fig. 2g).

7. Could the author explain why the sera from SARS-COV-2 S vaccinated individuals have 10-fold higher neutralization efficiency to T403R RaTG13 S than SARS-COV-2 S? There's only one point mutation on RaTG13 S. Could the authors compare the binding affinity between S or S variants and antibodies? Explain

Lower infectiousness of RaTG13 R403T S compared to the SARS-CoV-2 S might facilitate neutralization. In the initial experiments, we normalized for number and not the infectiousness of VSVpp. To address this issue, we analyzed a larger panel of sera from vaccinated individuals and normalized for infectivity (new Figs. 6d-f). Under these conditions CoV-2 and RaTG13 T403R S containing VSVpp are neutralized with similar efficiency.

8. The labels for S2 and S2' bands in Fig.S3 are upside down.

This figure has been omitted.

9. For Fig.S5 b, please check the labels for RaTG13 and its mutations, it's not corresponding to Fig. S5 a. Also, there's Fig.S5 c in the figure legend but no figure.

We thank the reviewer to making us aware of this and corrected the typo and the legend.

Reviewer #2: In the manuscript entitled "Spike mutation T403R allows bat coronavirus RaTG13 to use human ACE2" Zech and colleagues identified that the residue at position 403 in the spike protein of SARS-CoV-2 and the related bat-derived RaTG13 coronavirus plays an important role in regulating binding to the human ACE2 receptor.

Using reactive force field simulations authors showed that when an arginine (R) is present at position 403, as observed for SARS-CoV-2 spike, the spike-ACE2 interface is stabilized by charge interactions whereas when a threonine (T) is present, as is the case for RaTG13 spike, the interaction appears destabilized.

Infection of Caco-2 cells by VSV-based pseudotyped particles bearing wild-type (wt) or mutated SARS-CoV-2 spike protein showed that wild-type R403 favored infectivity while the R403T mutation decreased infection. Conversely, wt RaTG13 spike pseudotyped particles bearing T403 were associated with low infectivity, however the T403R mutation allowed for a substantial enhancement of infectivity while T403A mutation did not result in any increase. Similar results were obtained using a complementation assay based on a SARS-CoV-2 Δ S replicon system in HEK-293T cells expressing ACE2. The T403R mutation was also found to increase RaTG13 spike proteolytic processing in presence of human ACE2.

Infectivity findings using the Caco-2 cell line were further confirmed by infection of human gut organoids with VSV-based pseudotyped particles bearing SARS-CoV-2 and RaTG13 spike proteins. Using the VSV-based pseudotyping system, authors also tested ACE2 receptor usage from different bat species showing modest infectivity of wt SARS-CoV-2 spike pseudotyped particles in transfected cells expressing ACE2 from bats of the genus *Rhinolophus*. The T403R mutated RaTG13 spike was also found to be sensitive to EK-1 fusion inhibitor treatment as well as from sera of vaccinated individuals.

Overall, this is a well-conceived and executed study which brings to light new insights into the molecular determinants of ACE2 receptor binding for SARS-CoV-2 and the related bat coronavirus RaTG13. There are

several key points and figures that would require some clarification in particular the choice of cell types and drugs used in the study along with how some of the data are interpreted. Detailed comments and suggestions for improvements are found below.

We are pleased about the positive feedback.

Main comments

1. Throughout most of this work, intestinal Caco-2, HEK-293T, and gut organoids were used for infectivity assays with the exception of lung-derived Calu-3 cells which were used in supplementary fig. 4. While it is well-established that SARS-CoV-2 can productively infect enterocytes and cell types other than lung cells, can authors replicate findings obtained from Caco-2 and gut organoids in lung-derived cells, in particular infectivity assays? As coronavirus route of entry is often cell-type dependent, the choice of cell types used here should at least be discussed.

We used Caco-2 cells and gut organoids because they are highly susceptible to SARS-CoV-2 infection and because the gut might be an important target for secondary spread and pathogenesis. To address this valid concern, we confirmed the effect of the T403R change on RaTG13 Spike function in the human lung cancer cell lines Calu-3 and A549 (new Figs. 2b, 2c and Extended Data Fig. 1).

2. For fig. 4a, the titles “Pu ACE2” for *Rhinolophus affinis* ACE2 “Ma ACE2” and “Rh ACE2” are confusing—consider clarifying. Overall for fig. 4 it may be best to separate panels into two figures with panels a and b (receptor usage) forming one figure and c, d, and e (VSV pseudoparticle inhibition assays) forming another one as the relationship between these two sets of data are not explained clearly in the manuscript. Indeed, the title for fig. 4 only mentions human/bat ACE2 usage and the manuscript text describes fig. 4 in two separate paragraphs. In addition, currently the title for fig. 4—line 491 “Fig. 4: SARS-CoV-2 S and T403R RaTG13 S allow entry with human but not bat ACE2.” does not match what the data shown and with the statement in the manuscript text—lines 121-123: “Both WT SARS-CoV-2 S and (to a lesser extent) R403T SARS-CoV-2 S proteins were also capable of using bat (*Rhinolophus affinis*) ACE2 for viral entry although the overall infection rates were low (Fig. 4a). The figure title could mention that *Rhinolophus affinis* ACE2 allows some degree of infection that is lower than human ACE2.

*For clarity, we now provide the full names of the species of ACE2 origin. We agree that the infection and inhibition data were not well connected. To address this point and since neutralization analyses were expanded, we split the original figure 4 into two figures (now Fig. 5 and 6). We also made textual changes to address the point that *Rhinolophus affinis* ACE2 allows some albeit inefficient infection via the SARS-CoV-2 and RaTG13 T403R S proteins (lines 172-174).*

3. EK-1 is a peptide-based fusion inhibitor that targets the heptad repeat 1 (HR1) region within spike trimers. HR1 is located in the S2 fusion domain while position 403 is in the RBD region of the S1 domain, so the functional link between the two is unclear. This point is not discussed in the manuscript; however, this is important to interpret results appropriately. Could the authors provide more explanations as to the reasons why they chose this HR-targeting fusion inhibitor for testing the sensitivity of spike mutants at the distally-located position 403?

As now mentioned in the revised manuscript (lines 187-189) it has been previously shown that EK-1 has broad activity against various members of the coronavirus family (Xia et al, Science Advances, 2019). We included it as a positive control.

4. Supplementary fig. 2b clearly shows that the R403T mutant SARS-CoV-2 spike is incorporated much less efficiently into VSV particles than the wt spike. This begs the question of whether the 40% decrease in infectivity of SARS-CoV-2 R403T spike VSVpp compared to wt spike shown in fig. 2 is attributable to a modification in spike-ACE2 binding interface or simply because less mutated spike was incorporated into particles in the first place. This should be explained and discussed more explicitly in the manuscript.

Reviewer 2 is correct that the SARS CoV-2 R403T S was consistently expressed and found in moderately reduced levels in both pseudo-typed and according to our new data also replication-competent SARS-CoV-2. Thus, we agree that the modest attenuating effect of R403T on SARS-CoV-2 S might be due to reduced levels in the

particles and now clearly state this in the revised manuscript (lines 223-226). However, mutation of E37A reduced the efficiency of WT SCoV-2 S-mediated VSVpp infection to that observed for the R403T SCoV-2 S (new Fig. 3a). This indicates that the specific interaction between 403R in S and E37 in ACE2 contributes to the full infectious potential of SARS-CoV-2. Importantly, the T403R change enhanced RaTG13 S-mediated infection and ACE2 interaction without affecting its expression levels or pseudo-particle incorporation (new Figs 3d, 3e and 4b).

5. In the discussion, authors state—lines 163-165: “Notably, a positively charged residue at the corresponding position is present in the S proteins of the great majority of RaTG13-related bat coronaviruses (Supplementary Fig. 6)”. However, the data shown in supplementary fig. 6 for bat coronavirus sequences is confusing. In panel a, the sequence logo does not show “conservation of the RGD motif in bat coronavirus spike proteins” as stated in the figure title. The red box shows that it’s RGG that appears to be the most conserved motif. In panel b for bat coronavirus sequences the 3 residues in bold appears to be shifted one residue downstream and only 3 out of 9 sequences contain a basic residue that is stated in the text (R or K for BtCoV/Rp3, Bat SARS-like CoV, and Rhinolophus bat CoV) so not really a “a great majority”. In addition to the improvement suggested above, this supplemental figure would benefit from substantial revisions, for instance by using a uniform naming system for all sequences used in panel b (SARS-CoV-2 strains are named by accession numbers while the other sequences are names of viral strains or isolates) as well as residue numbers at least for the start and the end of the truncated sequences used in the analysis. Also, what is the rationale behind the choice of 9 bat coronavirus sequences in panel b? Do all these bat coronaviruses belong to Sarbecovirus subgenus, what is their phylogenetic relationship with RaTG13 and SARS-CoV-2? It is unclear why panel c was added to supplementary fig. 6.

As suggested, we considerably revised supplementary Figure 6 (now 7). The naming has been changed to the commonly used abbreviations (e.g. Lytras et al, Science, 2021) and corresponding accession numbers are provided. For the alignment bat/pangolin/human CoVs were chosen as representatives (compare Lytras et al, 2021, Science or Lam et al, Nature, 2020), the Sarbecoviruses are highlighted in the revised figure S7b. The sequence logo includes now only the ACE2-binding Spike Protein sequences of 50 representative bat, pangolin and human Sarbecoviruses and the conserved residue R403 is highlighted.

Minor comments

1. Lines 63-65: “We found that R403 is highly conserved in SARS-CoV-2 S proteins: only 233 of 1.7 million sequence records contain a conservative change of R403K and just 18 another residue or deletion.” Was the R403T mutation found among the 18 other residue changes identified? If so, it would be worth mentioning or discussing.

We have updated the numbers (lines 65-67). There is no clear trend towards another residue, nor are the numbers of non-conservative variations high enough to be concerning. Overall, a positive amino acid at position 403 is extremely high conserved (>99.9% of the available S sequences).

2. In fig. 2c, the representative brightfield images shown do not allow to see much difference between conditions, especially the reduced CPE for SARS-CoV-2 R403T compared to wt SARS-CoV-2. This figure panel would benefit from being presented more clearly.

This figure (now 2e) has been modified for clarity.

3. Some typos/mistakes identified:

- Line 118: “bat derived” the suggestion here is to add a hyphen: “bat-derived” *Done*
- Line 139: “SARS-Co-2” should be changed to “SARS-CoV-2” *Done*
- Line 161: “positive residue” should be changed to “positively charged residue” *Done*
- Line 203: “ACE2-bounded to SARS-CoV-2” should be changed to “ACE2 bound to SARS-CoV-2 spike” *Corrected.*
- Line 221: “H2O” the 2 should be lower case. *Done*
- Lines 232-233: “pBelo-SARSARS-CoV-2” ? SARSARS typo. *Corrected.*

- Line 468: The indicated arrows in fig. 1 appears grey not "purple". *Corrected in the figure legend.*

Reviewers' Comments:

Reviewer #1:

Remarks to the Author:

My questions have been addressed. The newly submitted manuscript didn't contain Fig. 1

Reviewer #2:

Remarks to the Author:

The authors have appropriately addressed all my comments and concerns, particularly concerning the issues of the cell types used in infectivity assays and clarifications on drug choice (EK-1) and presentation of figures. The results of the paper have been substantially bolstered and authors have carefully interpreted their new results. I commend the authors for their efforts in improving their work.

- In the figures of the merged PDF, fig. 1 is missing (instead there are two copies of fig. 2)
- For fig. 2 panel a, there appears to be an error in the labeling of one of the mutant spike proteins: "SCoV-2 S T403R" should be "SCoV-2 S R403T"

Reply to the reviewer`s comments (in *italic* letters)

Reviewer #1:

My questions have been addressed. The newly submitted manuscript didn't contain Fig. 1
We thank the reviewer for making us aware of this. Figure 1 is now included.

Reviewer #2:

The authors have appropriately addressed all my comments and concerns, particularly concerning the issues of the cell types used in infectivity assays and clarifications on drug choice (EK-1) and presentation of figures. The results of the paper have been substantially bolstered and authors have carefully interpreted their new results. I commend the authors for their efforts in improving their work.
We are pleased about the positive feedback.

In the figures of the merged PDF, fig. 1 is missing (instead there are two copies of fig. 2)
We thank the reviewer for making us aware of this. Figure 1 is now included.

For fig. 2 panel a, there appears to be an error in the labeling of one of the mutant spike proteins: "SCoV-2 S T403R" should be "SCoV-2 S R403T"
The labeling has been corrected.